# Variation of soil physicochemical properties of different vegetation restoration types on subtropical karst area in southern China

**Can Xiao[1], Ren You[2], Ninghua Zhu**  **[3,4,5]\*, Xiaoqin Mi[4], Lin Gao[4], Xiangshen Zhou[4], Guangyi Zhou[6]**

**1** Jiangxi Environmental Engineering Vocational College, Ganzhou, Jiangxi Province, China, **2** Faculty of Life Science and Technology, Central South University of Forestry and Technology, Changsha, Hunan Province, China, **3** Faculty of Forestry, Central South University of Forestry and Technology, Changsha, Hunan Province, China, **4** National Long-term Scientific Research Base for Comprehensive Control of Rocky Desertification in Wuling Mountain in Hunan Province, Jishou, China, **5** National Long-Term Scientific Research Base of Mid subtropical Forestry, China, **6** Institute of Tropical Forestry, CAF, Guangzhou, China

\* zhuninghua@yahoo.com

**Data Availability Statement:** All relevant data are within the paper and its Supporting Information files.

**Funding:** The following grant information was disclosed by the authors: Central Public-interest

## Abstract

To carry out differentiated ecological restoration activities and formulate appropriate environmental conservation strategies for karst regions, it is essential to investigate the impact of ecological restoration and forest management strategy differences on soil properties. The karst region in Xiangxi, Hunan province, China was selected as the study site. Here, we determined soil physical and chemical differences in soil profiles of karst areas with ecological restoration activities. The results showed that (1) the soil properties showed a significant difference between the restoration vegetation and uncultivated land, especially in soil physical properties. The soil moisture conversion coefficient (83.0%) and soil bulk density (1.37g/cm$^3$) of *Liriodendron chinense* (Hemsl.) Sarg reached the highest value among 12 vegetations. 2) The topsoil was more sensitive to ecological restoration. Soil physical properties in the topsoil samples from the forest management areas were significantly higher than uncultivated lands ($P < 0.05$). (3) Redundancy analysis showed that the soil chemical content differed significantly among the types of forest vegetation restoration and different soil layers. Among the nutrients analysis, Mg, Zn and K were the main factors affecting soil properties in the rocky desertification areas. Therefore, our results recommend planting the broadleaved deciduous forest as the preferred forest among three different forest types to enhance soil fertility and water conservation functions, especially in subtropical karst areas ecosystems, which provided for making scientific forest restoration management in the karst region.

## Introduction

More than two billion ha have been identified globally as potentially suitable for either passive or active forest restoration [1]. Most restoration projects have focused on the recovery of

Scientific Institution Basal Research Fund: CAFYBB2019SZ003 and Hunan Province Forestry Science and Technology Research and Innovation Project: XLKY2023-30. The funders had no role in study design, data collection and analysis, decision to publish, or preparation of the manuscript.

**Competing interests:** The authors have declared that no competing interests exist.

vegetation to assess restoration success and made a great process [2]. For the past three decades, to prevent soil erosion and desertification and improve water conservation capacity, the Grain to Green Program (GTGP) has been implemented by the Chinese government [3]. Forest in the process of recovery has concentrated mainly on vegetation structure, species diversity and ecosystem processes [4], ecosystem productivity [5], and susceptibility to invasions [6, 7]. These mechanism factors are relatively clear [8–10], but the feedback relationships between plants and soil, and the succession processes and regulation mechanisms of plant communities still remain unclear [11]. Yet, knowledge of such feedback relationships is urgently required to guide instrumental in predicting future scenarios under varying environmental conditions and designing measures for vegetation restoration at different succession stages [12].

Large-scale afforestation increased ground cover and caused changes in soil physical and chemical properties. The interaction between soil and vegetation indicates that they always co-evolved and developed, which are viewed as an important mechanism for forest succession and development [13]. The association between soil and above-ground vegetation may change the course of restoration constantly [14]. Previous studies have concluded that Forests as ecosystem engineers not only have species-specific on effects the soil environment (soil animal communities and soil microbial communities) [15], including soil nutrients, moisture, and structure [16] of plants but also significantly influence plant diversity and ecosystem productivity [17]. Previous studies on soil elements during forest restoration have focused on major elements, but the cycling and feedback effects of mineral nutrients between above- and below-ground forest ecosystems are complex [18]. Therefore, the study of differences in soil physicochemical properties, especially trace metal elements, among different vegetation restoration types is an important guideline for improving ecological restoration of natural and planted forests, especially in the subtropical karst region of southern China.

Karst is a distinctive topography, created by the action of acidic water on carbonate bedrock, such as limestone, dolomite, or marble [19], which is mainly distributed in southwest China which is one of the three major continuous karst areas in the world [20]. Due to its specific geologic and climate conditions, the karst area is characterized by small environment capacity, weak anti-disturbance, low stability and powerless self-adjustment [13]. However, the long-term severe human disturbance has a serious effect on subtropical karst region vegetation restoration, with a complex topography and climate change resulting in forest restructuring and a decrease in the functioning of an ecological security barrier.

The karst region of southwest China covers an area of 550,000 km$^2$ [21], which is one of the main regions involved in the Grain to Green Program (GTGP). These abandoned lands are undergoing a transition from crops to forests and other secondary vegetation, with changes in ecosystem structure, processes and functions [22]. Since the 1990s, government policies have forced farmers to give up some land in karst areas where erosion losses are particularly high. As the project is implemented, the agricultural and stony lands are gradually restored to grasslands, shrubs and forests, depending on the time of abandonment.

In the present study, soil nutrient characteristics were investigated in the plantation soil of three different types (CF, EB, BD) of forest vegetation (coniferous, evergreen broadleaved, broadleaved deciduous) which includes 12 different tree species underlaid, which represent the main artificial forests of vegetation restoration in the subtropics of China. Our objective was to investigate how soil physicochemical properties and vegetation features change and how soil and vegetation stimulate vegetation restoration individually and collectively. We formulated two hypotheses: (1) that soil elements content of different tree species showed significant differences; and (2) that vegetation restoration would have an obvious positive effect on soil physicochemical properties and vegetation features. This analysis and evaluation served as

a guide to the vegetation restoration and protection of karst region ecosystems in southern China.

## Materials and methods

### Study site

The research area is located in Qingping town, Yongshun county, Xiangxi Tujia and Miao Autonomous Prefecture, Hunan province. The geographical location is 29˚3'N, 110˚13'E. The topography features a typical low hilly landscape, at an altitude of 320–820 m above sea level, which belongs to the central area of Wuling Mountain. The climate is characterized by southeast monsoon and a mid-subtropical humid climate with an annual average precipitation of 1300–1500 mm (primarily between April and August) and an annual mean air temperature of 15.8–16.9˚C, and the slope is 23˚. The rock exposure is 60%, and which parent rock is limestone, which belongs to the area of severe rocky desertification. The shrubs growing in the study area during vegetation restoration are *Camellia sinensis* (L.) O. Ktze., *Eriobotrya deflexa*, *Lindera glauca*, *Alangium chinense*, *Siegesbeckia orientalis* L.,. The main herbs are *Duchesnea indica* (Andr.), *Cyperus rotundus* L. and *Kalimeris indica* (L.) Sch. Bip., *Tripterospermum chinense* (Migo) H. Smith.

### Soil sample collection

A space-for-time substitution approach was used to collect soil profile samples from the core area and buffer zone of the Xiangxi Tujia and Miao Autonomous Prefecture Ecological Research and Experimental Station in January 2019 (Fig 1). In the study area, select 12 representative native precious tree species and a control group of wasteland (natural succession land) (Table 1 & Fig 2). The soil profiles were examined by digging all the way to the bedrock after removing surface roots. From bottom to top, one sample was collected every 15cm, and each sample was mixed with 3 soil samples collected from the same horizontal plane. The field plot design was carried out in geographical environment characteristics of the tree species, three plots were set respectively, with a plot area of 20m×20m. Five points were selected from each sample plot using the quincunx sampling method, and mixed soil samples from the 0–15 cm depth were collected. The characteristics of the sample sites and the surrounding environmental information were recorded, primarily longitude and latitude, altitude, geomorphology, vegetation type, bedrock exposed degree, and disturbance degree. Gravel, animal and plant residue were removed from the soil samples. Then, the samples were air-dried, ground, sieved through 2 mm, 1 mm and 0.25-mm sieves, and bottled for use in subsequent analyses.

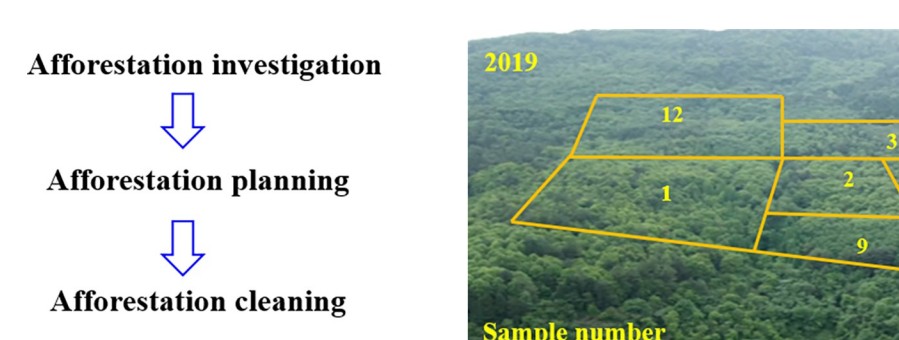

**Fig 1. The basic information of the experimental plot.**

**Table 1. Study area information of different vegetation types.**

| Vegetation restoration types | Dominant species | Afforestation time (yr) | Altitude | Degree of rocky desertification | Under-forest shrub and herb species |
|---|---|---|---|---|---|
| Broadleaved deciduous | *Kalopanax septemlobus* (Thunb.) Koidz. | 1979 | 513 | Moderate | *Cryptotaenia japonica* Hassk., *Smilax china* L., *Achyranthes bidentata* Blume, Dryopteris microlepis (Bak.) C. Chr., *Lysimachia christinae* Hance |
| | *Corylus chinensis Franch.* | 1977 | 510 | Moderate | *Cyclosorus interruptus* (Willd.) H. Ito, *Lonicera japonica* Thunb., *Aster tataricus L. f.*, *Ophiopogon bodinieri* Levl., *Smilax china* L., *Hedera nepalensis var. sinensis* (Tobl.) Rehd. |
| | *Nyssa sinensis* Oliv. | 1979 | 540 | Intense | Makino, *Cyclosorus interruptus* (Willd.) H. Ito, *Rubus corchorifolius* L. f., *Camellia oleifera* Abel., Smilax china L, *Rosa* sp. |
| | *Quercus acutissima* Carruth. | 2002 | 551.8 | Moderate | *Cyclosorus interruptus* (Willd.) H. Ito, Cyperus rotundus L., *Semiaquilegia adoxoides* (DC.) Makino, *Lindera glauca* (Sieb. et Zucc.) Bl |
| | *Liriodendron chinense* (Hemsl.) Sarg | 1982 | 467 | Intense | *Camellia oleifera* Abel. (Willd.) H. Ito, *Zanthoxylum armatum DC.*, Cyperus rotundus L., *Lysimachia christinae* Hance |
| | *Choerospondias axillaris* (Roxb.) Burtt et Hill. | 1984 | 421.5 | Intense | Cyperus rotundus L., *Semiaquilegia adoxoides* (DC.) Makino, *Aster tataricus L. f.*, *Rubus corchorifolius* L. f., *Rosa* sp., *Camellia japonica* L. |
| Coniferous | *Cupressus funebris* Endl. | 1995 | 460 | Moderate | *Amphicarpaea trisperma* Baker, *Agrimonia pilosa* Ldb., *Rubia cordifolia* L. |
| | *Metasequoia glyptostroboides* Hu & W. C. Cheng | 1979 | 451.5 | Moderate | *Semiaquilegia adoxoides* (DC.) Makino, *Akebia trifoliata (Thunb.)* Koidz., *Solanum pseudocapsicum* L., *Agrimonia pilosa* Ldb. |
| | Taiwania cryptomerioides | 1979 | 558.9 | Moderate | *Cyclosorus interruptus* (Willd.) H. Ito, Cyperus rotundus L., *Rosa* sp., Serissa japonica(Thunb.)Thunb., *Cryptotaenia japonica* Hassk. |
| Evergreen broadleaved | *Cinnamomum septentrionale* Hand.-Mazz | 1985 | 450.4 | Moderate | *Lonicera japonica* Thunb., *Cyclosorus interruptus* (Willd.) H. Ito, *Solanum nigrum* L., *Agrimonia pilosa* Ldb. |
| | *Lindera megaphylla* Hemsl. | 1987 | 470 | Moderate | *Rubus corchorifolius* L. f., *Camellia oleifera* Abel., Smilax china L, *Rosa* sp., *Agrimonia pilosa* Ldb., Cyperus rotundus L. |
| | Michelia maudiae Dunn | 1986 | 463 | Moderate | *Lonicera japonica* Thunb., *Aster tataricus L. f.*, *Ophiopogon bodinieri* Levl., |
| – | uncultivated land | – | 443 | Intense | Cyperus rotundus L., Clinopodium chinense (Benth.) O. Ktze., *Semiaquilegia adoxoides* (DC.) Makino, *Aster tataricus L. f.*, *Smilax china* L |

## Experimental methods

In this manuscript, we analyzed the soil samples according to the methods in "Soil Physical and Chemical Analysis" [23]. Soil physical indicators such as Soil density, field water capacity, and capillary density were determined using oven drying and the ring-knife method [24–26]. The chemical properties required for the experiment in this study include: Al, Ca, Cd, Cu, Fe, K, Mg, Mn, Na, P, Pb, and Zn. Soil samples were decomposed by the HCL-HNO$_3$ leaching method [27]. Weigh 2.0g of air-dried soil sample in a triangular flask, then add 15 ml HCL (1 volume of HCL + 1 volume of distilled water) and 5 ml HNO$_3$, add the plug and shake for 30 min, filter, fix the volume to 100ml, and waiting for measurement. A method to determine soil exchangeable potassium (K), calcium (Ca), sodium (Na), and magnesium (Mg) by using inductively coupled plasma (ICP-OES) (Plasma2000) and extraction with ammonium acetate was developed [28], which are measured by the Research Institute of Tropical Forestry, Chinese Academy of Forestry. The contents of P, Fe, Cu, Mn, Pb, Cd and Zn in soil were determined by ICP-AES.

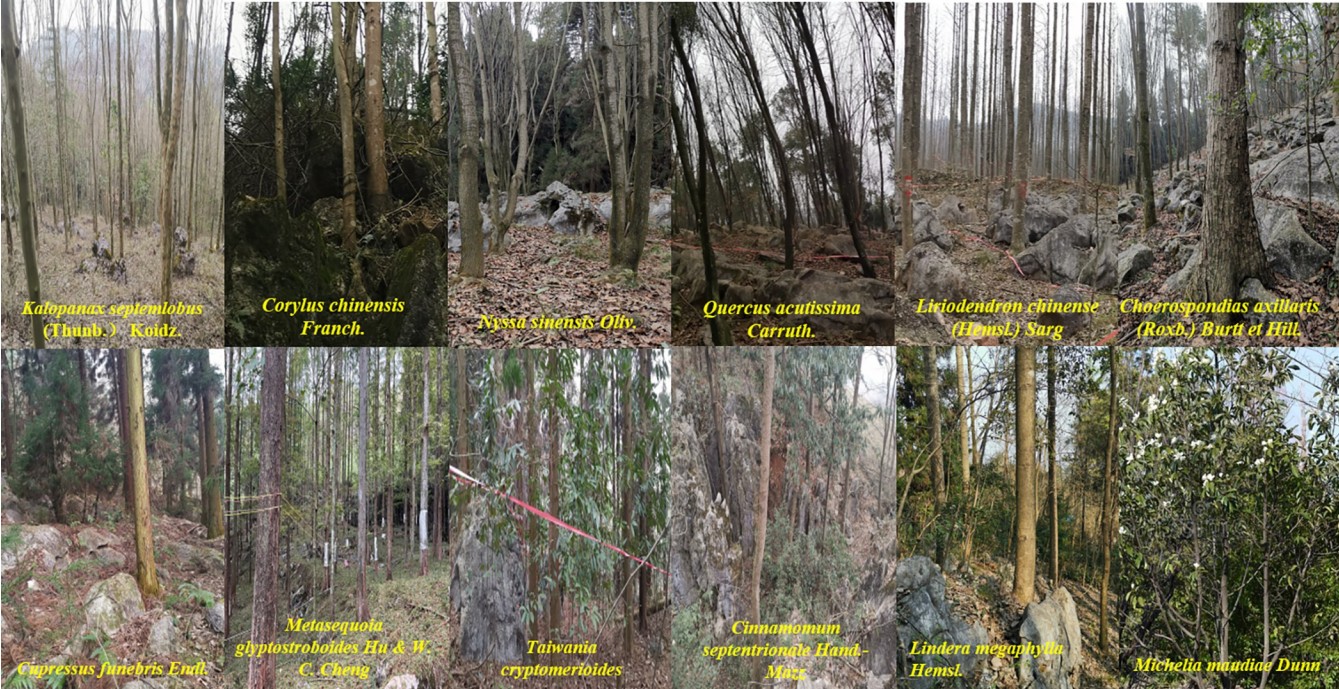

**Fig 2. The photos of the 12 representative native precious tree species.**

## Data processing

The species diversity index was described by the Shannon-Wiener diversity index, Simpson diversity index and Pielou evenness index. The calculation process is based on the vegan package in R. The calculation method is as follows:

$$\text{Species richness index}: D = \frac{S}{\ln A}$$

$$\text{Shannon}-\text{Wiener index}: H' = -\sum_i p_i \log_b p_i,$$

$$\text{Simpson diversity index}: 1 - \sum p_i^2,$$

$$\text{Pielou evenness index}: J = \frac{H'}{\ln S'},$$

$p_i = N_i/N$, where $N_i$ is the number of individuals of the $i$ th species, $i$ and $b$ is the base of the logarithm, $N$ is the sum of the number of individuals of all species and $S$ is the number of species.

In this study, soil physical and chemical properties of 12 tree species were treated on average (groups of repetitions), and then 12 different tree species were divided into three different forest vegetation types (CF, EB, BD) for further data processing.

One-way analysis of variance (ANOVA) and the Tukey-HSD test were used to analyze differences in soil physical properties between different vegetation types and different soil layers. Cluster analysis was used to classify the chemical properties of different vegetation types and soil layers. Redundancy analysis was used to determine the relationship between soil physical and chemical properties in different soil layers of different vegetation restoration types. All

data were processed by Excel software, and statistical analysis was performed using R software [29].

# Results

## Differences of soil physical properties among different vegetation restoration

The soil physical properties during vegetation restoration are presented in Fig 3. The uncultivated land showed no difference with CF, BD and EB in soil bulk density, but showed a significant difference with other vegetation types in soil moisture conversion coefficient ($K_s$), total porosity, and soil water content. Among 12 native species, Liriodendron chinensis reached the highest value, followed by uncultivated land but showed no difference between different soil layers (Fig 3 & Fig 4).

The correlation analysis of soil physical properties for the 12 different species showed an extremely significantly negative correlation between $K_s$ and soil water content (SOC) ($p<0.01$). $K_s$ has an extremely significantly negative correlation with other physical properties except shown the positive correlation with soil bulk density (SBD) (correlation coefficient = 0.44). SOC has an extremely significantly positive correlation with other physical properties except shown the negative correlation with SBD (correlation coefficient = -0.48). SBD has an extremely significantly negative correlation with other physical properties except for soil air-filled porosity (SAP) (correlation coefficient = -0.05). Maximum water-holding capacity (MWC) was negatively correlated with capillary water-holding capacity (CWC), minimum field capacity (MFC), SAP, capillary porosity (CP), and total porosity (TP) content ($p<0.01$). CWC was not significantly positively correlated with SAP ($p>0.05$). MFC was significantly positively correlated with TP ($p<0.05$) and not significantly positively correlated with SAP and CP ($p>0.01$). There was an extremely significant positive correlation between SAP and CP content ($p<0.01$). CP was significantly positively correlated with TP (P<0.05) (Table 2).

## Hierarchical cluster analysis for soil nutrient elements indices

The Euclidean distance-based hierarchical clustering organized the 12 tree species into three key subclusters: clusters I, II and III.

The cluster analysis results are robust to different clustering methods (whether hierarchical or partitioning methods) and to different specifications of the same method (whether the k-means or k-medians algorithm in the case of partitioning methods). Fig 5 showed the composition of the clusters resulting from analysis through a k-means algorithm. Cluster 1 comprises three soil layers (0-15cm, 15-30cm, >30cm) of *Cinnamomum septentrionale* and *Cupressus funebris* also including *Liriodendron chinense* (Hemsl.) Sarg (0-15cm and 15-30cm). This group reflects the highest degree of both inner cohesion and separation from the other groups, as shown by the largest value of the silhouette width. Cluster 2 is composed of *Liriodendron chinense* (>30cm) and *Choerospondias axillaris* (>30cm), whereas cluster 3 is the largest group and includes the following components: *Metasequoia glyptostroboides*, *Cupressus funebris*, uncultivated land, Taiwania cryptomerioides, *Nyssa sinensis* Oliv., *Kalopanax septemlobus*, *Quercus acutissima*, *Choerospondias axillaris* (0-15cm, 15-30cm).

## Effects of stand type characteristics on soil basic physical and chemical properties

In Fig 6, the result of RDA demonstrated there were differences in soil physical and chemical properties in different soil layers of different vegetation restoration types in karst areas. The

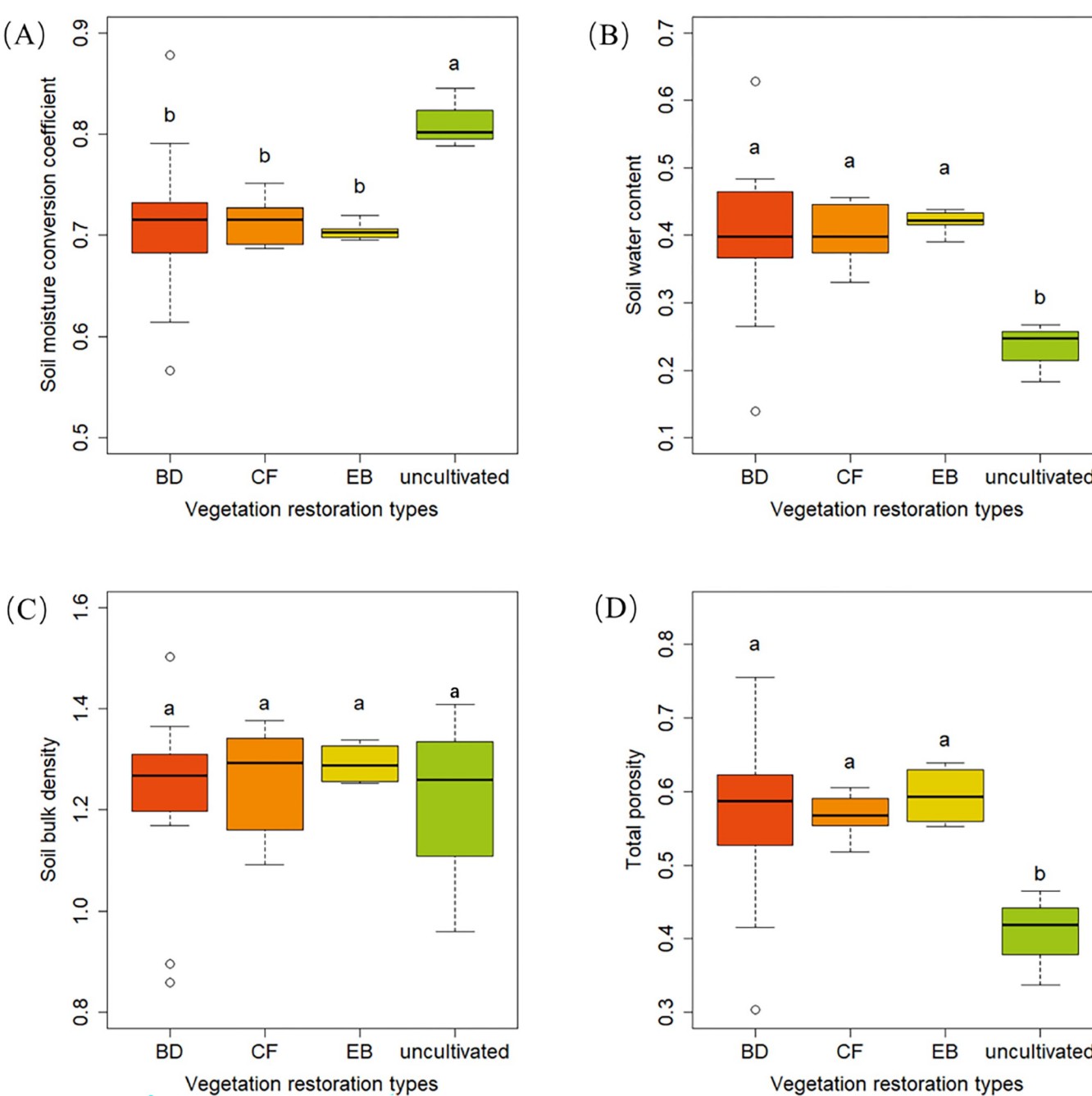

**Fig 3. Differences in soil physical properties among different vegetation types.**

total explained variances were the highest for the 0-15cm soil layer, which reached 92.98%. The first ordination axis accounted for 66.00% of the fitted variation. For the 15–30 cm soil layer, the first principal component axis and the second principal component axis explained 30.36% and 19.93%, respectively. For the >30cm soil layer, the first principal component axis and the second principal component axis explained 52.31% and 17.44%, respectively. Mg, K, Zn had significant effects on different soil layer properties. We can also find the relationship between the vegetation and chemical elements in different soil layers in Fig 6. For 0–15 cm soil

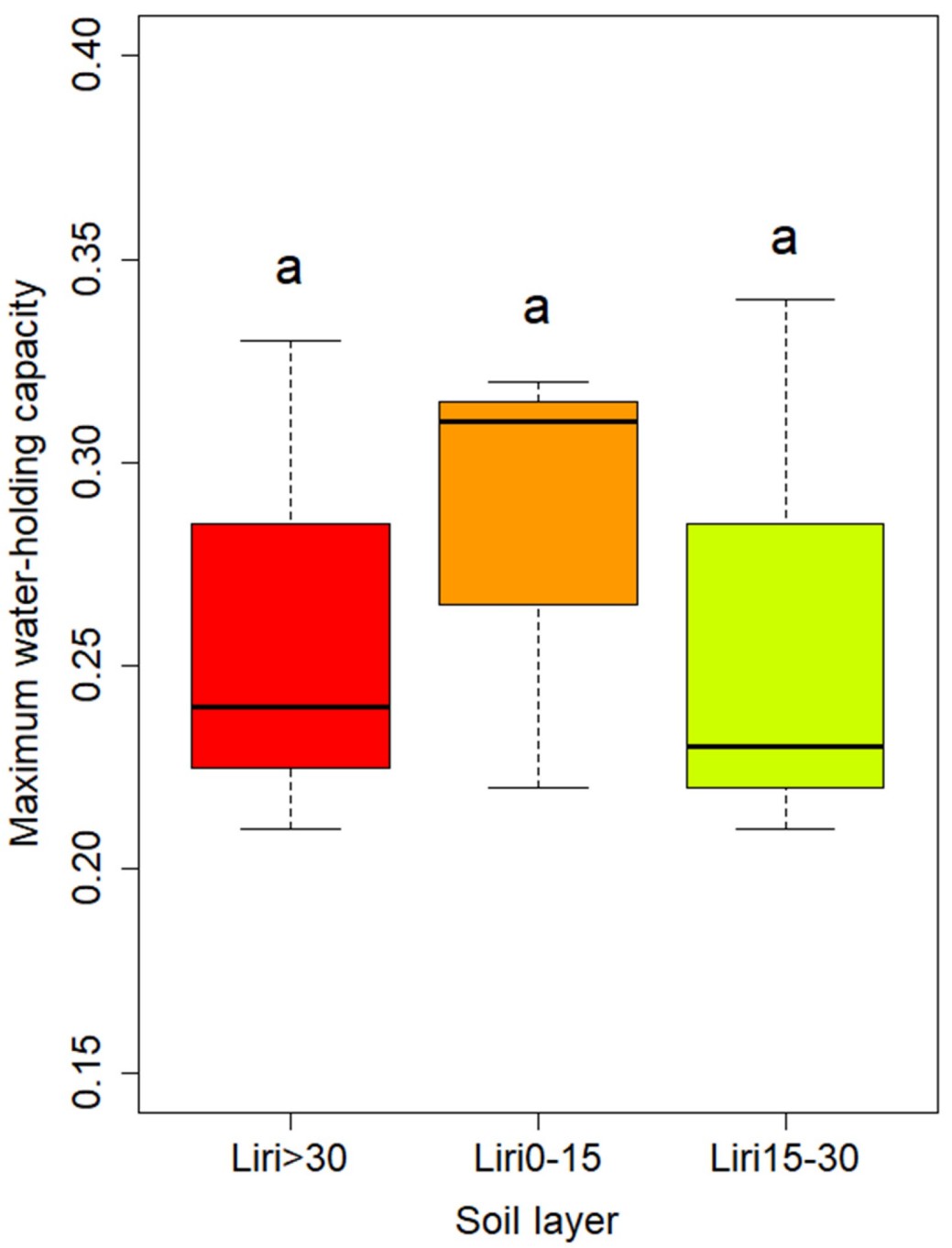

**Fig 4. Liriodendron chinensis maximum water-holding capacity in different soil layers.**

layer, the BD showed the positive relation with chemical elements especially Ca. And the EB showed the negative relation with chemical elements. For 15–30 cm soil layer, the BD showed the negative relation with Na and showed the strongly positive relation with Ca and Cd. The EB showed no relation with Na. For >30 cm soil layer, the BD showed the negative relation with chemical elements. The EB shown the strongly positive relation with Ca, Cd and Pb.

**Table 2. Correlation between soil physical properties.**

|  | $K_s$ | SOC | SBD | MWC | CWC | MFC | SAP | CP | TP |
|---|---|---|---|---|---|---|---|---|---|
| $K_s$ | 1.00 |  |  |  |  |  |  |  |  |
| SOC | -0.99*** | 1.00 |  |  |  |  |  |  |  |
| BD | 0.44*** | -0.48*** | 1.00 |  |  |  |  |  |  |
| MWC | -0.84*** | 0.85*** | -0.73*** | 1.00 |  |  |  |  |  |
| CWC | -0.85*** | 0.87*** | -0.76*** | 0.94*** | 1.00 |  |  |  |  |
| MFC | -0.86*** | 0.87*** | -0.69*** | 0.91*** | 0.96*** | 1.00 |  |  |  |
| SAP | -0.27** | 0.25** | -0.05 | 0.45*** | 0.13 | 0.15 | 1.00 |  |  |
| CP | -0.89*** | 0.86*** | -0.40*** | 0.81*** | 0.88*** | 0.87 | 0.13 | 1.00 |  |
| TP | -0.85*** | 0.82*** | -0.34** | 0.87*** | 0.77*** | 0.78*** | 0.60*** | 0.87*** | 1.00 |

*$K_s$*: soil moisture conversion coefficient (%); *SOC*: soil water content; *SBD*: soil bulk density; *MWC*: maximum water-holding capacity; *CWC*: capillary water holding capacity; *MFC*: minimum field capacity; *SAP*: soil air-filled porosity; *CP*: capillary porosity; *TP*: total porosity

** Correlation is significant when the confidence (double test) is 0.01.

* Correlation is significant when the confidence (double test) is 0.05.

## Discussion

### Soil chemical-physical properties during vegetation restoration in different ecosystems

Soil is the result of the comprehensive action of topography, climate, biology, parent material and time and changes with vegetative succession [30]. The water storage capacity of the soil is affected by the soil physical and chemical properties [31]. Soil bulk density and soil capacity have significant effects on hydrological processes, which are crucial in the supply and storage of water, nutrients, and oxygen in the soil [32, 33]. The size of soil porosity plays an important role in quantifying soil structure, which can influence soil hydraulic conductivity, solute convection and water retention [34]. Therefore, these indicators can be used to evaluate the impact of vegetation restoration on soil properties [35]. Soil's physical properties are different in different ecosystems. Zhang et al found in the central part of the Loess Plateau that soil texture, porosity and bulk density were the key factors affecting soil water holding capacity and soil water availability [36]. There is research estimated the status of soil carbon amounts after revegetation with trees and grass in South West Iceland (Hafnarmelar), suggested that where land has been properly restored or kept in natural condition, soil properties improve significantly especially when trees are part of the restored vegetation [37]. However, the soil physical properties of karst landform areas after vegetation restoration have not been thoroughly studied. Our results showed that soil $K_s$ decreased, and the contents of SBD, MWC, CWC, MFC, SAP, CP, and TP increased compared to the uncultivated sample area except for *Liriodendron chinense* (Hemsl.) Sarg (Fig 2), indicating that soil physical properties improved significantly. These results are partially consistent with our hypothesis and with the results of Zhang et al. [38]. After agricultural abandonment and the restoration of natural vegetation, soil nutrient sources are mainly composed of litter and plant roots. The SOC, TP, TK contents in the soil following post-agricultural succession were significantly higher than those in the uncultivated sample area, indicating that the restoration of natural vegetation improved SOC sequestration and nutrient accumulation [39].

In our study, the results show that chemical elements had significant effects on vegetation. Vegetation cover can have significant effects on soil properties [40, 41], primarily due to its input of organic matter to the soil via the supply of carbon and energy sources from root

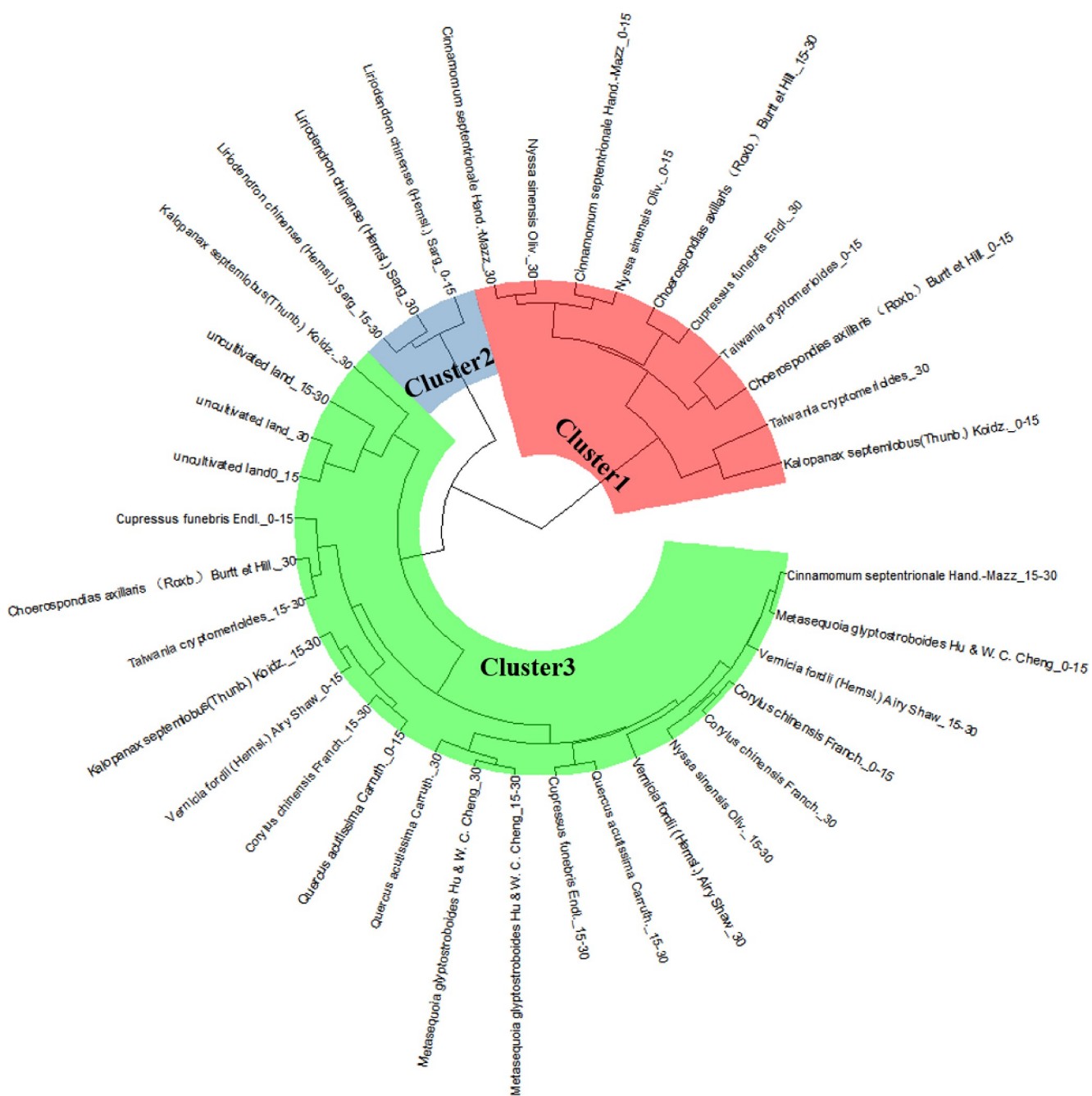

**Fig 5. Cluster analysis of chemical properties of different vegetation types.**

exudates and plant remains. In a comparative study of the concentrations of ten nutrients in 83 (mostly herbaceous) species from central England, only Ca (positively) and Mn (negatively) were consistently correlated with soil pH [42].

## Plant diversity in different forest management strategies in karst region

A previous global meta-analysis that comprised different terrestrial and aquatic ecosystems to those presented here indicated that restoration of degraded systems enhanced overall

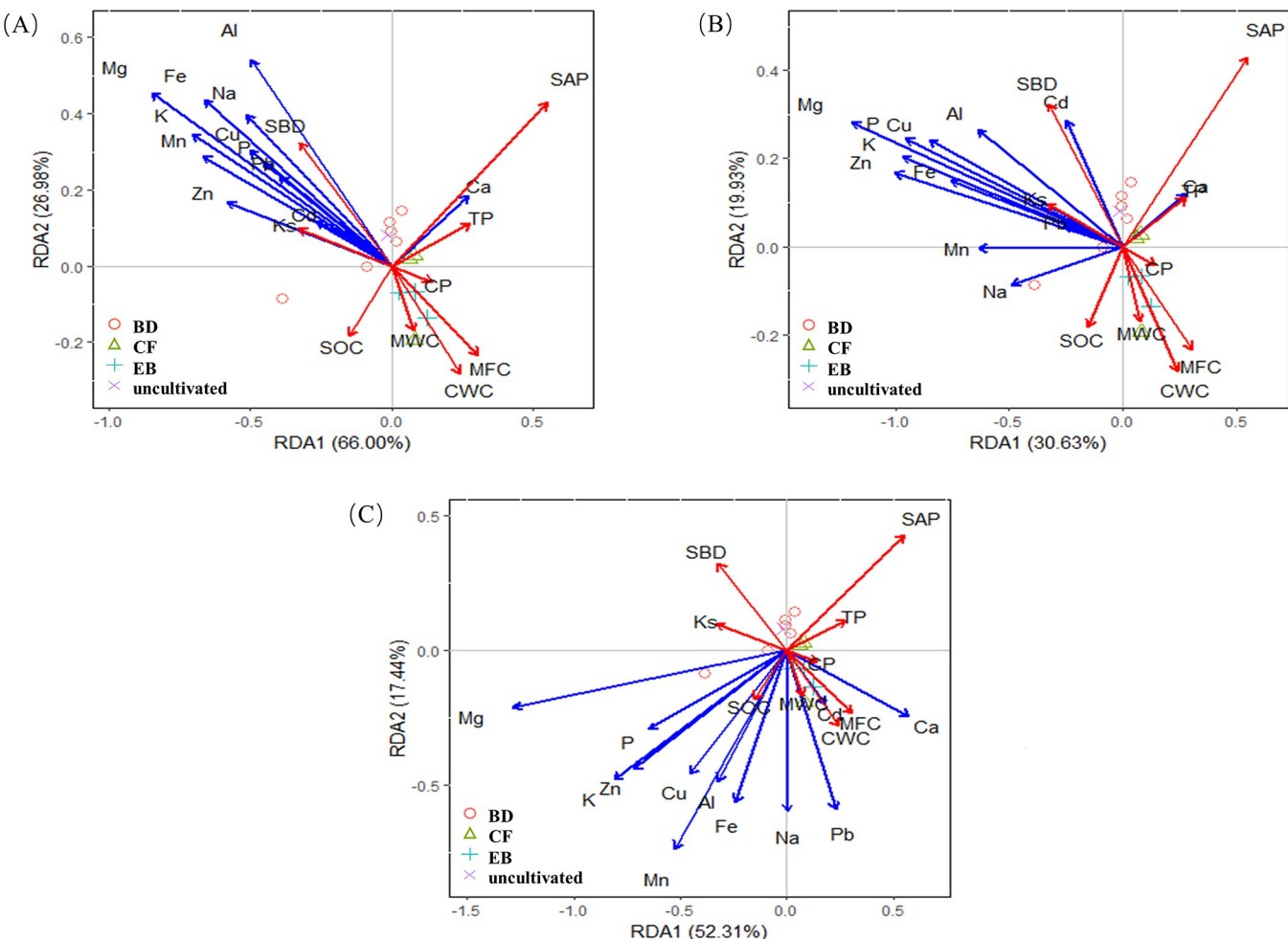

**Fig 6. Redundancy analysis (RDA) results show the relationship between soil physical and chemical properties of species in different soil layers.** Red and blue arrows reflect the soil physical properties and chemical properties of the RDA1 and RDA2 axis, respectively. Different vegetation types are represented by different symbols.

biodiversity by 44% [43]. In rocky desertification areas, the soil layer is barren, the nutrient content is low, and the composition of plant community is relatively single due to soil erosion [44]. In a karst ecosystem in China, the species diversity rose steadily with community succession, where the vine-shrub community had the most abundant species, and the highest α-diversity index was detected in secondary forests, and then the diversity decreased in the old-growth evergreen broad-leaved forest. In our study, the tree biomass is higher than in mixed evergreen deciduous broadleaf forest in Southwest in China [45], and also higher than in the same type of karst forest in the neighboring Guizhou province [46]. The difference may be attributed to differences in the plot areas studied. We also found that the karst forest had lower AGB than typical forests in non-karst regions in the same climate zone. Plant diversity generally increases with succession and tends to be exceptionally high in successional communities, while it tends to be exceptionally low in old field ecosystems when there is strong species dominance.

The Species richness index, Shannon-Wiener index, Simpson's diversity index and Pielou index of the herb layer were calculated. The result showed that the species richness index, Shannon-Wiener index, Simpson's diversity index and Pielou index of sample area which had

**Table 3. Plant diversity indexes of herbs layer of different vegetation types.**

| Species | Diversity index (herb layer) | | | |
|---|---|---|---|---|
| | Species richness (S) | Simpson (D) | Shannon-Winner (H') | Species evenness(E) |
| Liriodendron chinense | 4 | 0.58 | 0.95 | 0.772 |
| *Metasequoia glyptostroboides* | 6 | 0.30 | 0.68 | 0.147 |
| Michelia maudiae Dunn | 3 | 0.93 | 0.07 | 0.0035 |
| *Nyssa sinensis* | 5 | 0.50 | 0.84 | 0.770 |
| Taiwania cryptomerioides | 3 | 0.24 | 0.36 | 0.113 |
| *Choerospondias axillaris* | 4 | 0.47 | 0.81 | 0.401 |
| *Corylus chinensis Franch.* | 7 | 0.62 | 1.06 | 1.255 |
| *Cupressus funebris* | 7 | 0.62 | 1.06 | 1.255 |
| *Quercus acutissima* | 3 | 0.45 | 0.77 | 0.483 |
| *Kalopanax septemlobus* | 3 | 0.39 | 0.60 | 0.525 |
| *Lindera megaphylla* | 3 | 0.39 | 0.60 | 0.525 |

been made forest restoration managements had significant increase than uncultivated area. The Shannon-Wiener index of *Metasequoia glyptostroboides* Hu & W. C. Cheng showed the highest among these tree species. The Simpson's diversity index of *Cinnamomum septentrionale* showed the highest among these tree species, but the Pielou index of *Cinnamomum septentrionale* reached the lowest value (Table 3 and Table 4).

## Conclusion

The soil chemical and physical properties and plant diversity of 12 vegetation types in Northwest Hunan were analyzed. The result showed that the soil properties showed a significant difference between the vegetation and uncultivated land ($p<0.05$). The soil moisture conversion coefficient (83.0%) and soil bulk density (1.37g/cm$^3$) of *Liriodendron chinense* (Hemsl.) Sarg reached the highest value among 12 vegetations. For these characteristics, the broadleaved deciduous forest may be the most suitable forest type in forest restoration management in the subtropical karst area of southern China. The influence of human factors was significant. There were obvious differences in vegetation community composition among the different rocky desertification areas in Southwest Hunan. The chemical properties also showed a significant difference between vegetation and uncultivated land via cluster analysis. The results of

**Table 4. Plant diversity indexes of bush layer of different vegetation types.**

| Species | Diversity index (bush layer) | | | |
|---|---|---|---|---|
| | Species richness (S) | Simpson (D) | Shannon-Winner (H') | Species evenness(E) |
| Liriodendron chinense | 1 | 0 | 0 | 0 |
| *Metasequoia glyptostroboides* | 1 | 0 | 0 | 0 |
| Michelia maudiae Dunn | 1 | 0.14 | 0.276 | 0.121 |
| *Nyssa sinensis* | – | – | – | – |
| Taiwania cryptomerioides | 2 | 0.50 | 0.69 | 0.351 |
| *Choerospondias axillaris* | 3 | 0.44 | 0.64 | 0.255 |
| *Corylus chinensis Franch.* | – | – | – | – |
| *Cupressus funebris* | 2 | 0.50 | 0.69 | 0.351 |
| *Quercus acutissima* | 3 | 0.44 | 0.64 | 0.255 |
| *Kalopanax septemlobus* | 2 | 0.49 | 0.82 | 0.407 |
| *Lindera megaphylla* | 2 | 0.49 | 0.82 | 0.407 |

redundancy analysis show that Mg, K, Zn had significant effects on different soil layer properties. There were differences in soil physical and chemical properties in different soil layers of different vegetation restoration types in karst areas. The abilities of vegetation types to adapt to the rocky desertification environment were significantly different under different forest restoration management.

This study expounded on the relationship between vegetation types, soil and plant diversity, and the results were of great significance for making scientific forest restoration management. In the process of vegetation restoration, suitable species should be selected to restore ecology according to different rocky desertification degree and its characteristics.

## Supporting information

**S1 File.**
(XLS)

## Acknowledgments

The authors should thank the National long-term scientific research base for the comprehensive control of rocky desertification in Wuling Mountain of Jishou City, Hunan province and Research Institute of Tropical Forestry, CAF for their supporting the fieldwork and thank for Guangyi Zhou for their assistance in measuring the soil physical and chemical properties.

## Author Contributions

**Conceptualization:** Xiaoqin Mi.

**Data curation:** Ren You, Ninghua Zhu.

**Formal analysis:** Can Xiao, Guangyi Zhou.

**Funding acquisition:** Ninghua Zhu, Guangyi Zhou.

**Investigation:** Xiaoqin Mi.

**Methodology:** Can Xiao, Ninghua Zhu, Xiaoqin Mi, Lin Gao, Xiangshen Zhou, Guangyi Zhou.

**Resources:** Ninghua Zhu.

**Software:** Ren You, Xiangshen Zhou, Guangyi Zhou.

**Supervision:** Lin Gao, Xiangshen Zhou.

**Visualization:** Lin Gao.

**Writing – original draft:** Ren You.

**Writing – review & editing:** Can Xiao, Ren You.

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
