## [Decision Letter · Decision Letter 0]

1 Dec 2022

PONE-D-22-29404Variation of soil physicochemical properties of different vegetation restoration types on subtropical karst area in southern ChinaPLOS ONE

Dear Dr. Ninghua,

Thank you for submitting your manuscript to PLOS ONE. After careful consideration, we feel that it has merit but does not fully meet PLOS ONE’s publication criteria as it currently stands. Therefore, we invite you to submit a revised version of the manuscript that addresses the points raised during the review process.

The study has value but the manuscript has some problems as suggested by the reviewers. The authors should respond to the comments of the reviewers one by one and revise the manuscript accordingly. The revised manuscript might be sent to the reviewers for further reviewing.

We look forward to receiving your revised manuscript.

Kind regards,

Jian Liu

Academic Editor

PLOS ONE

Journal Requirements:

"NO authors have competing interests"

7. Please include a separate caption for each figure in your manuscript.

8. Please ensure that you refer to Figure 7 in your text as, if accepted, production will need this reference to link the reader to the figure.

9. Please upload a new copy of Figure 5 as the detail is not clear. Please follow the link for more information:

https://blogs.plos.org/plos/2019/06/looking-good-tips-for-creating-your-plos-figures-graphics/

https://blogs.plos.org/plos/2019/06/looking-good-tips-for-creating-your-plos-figures-graphics/

10. We note that Figure 1 in your submission contain map images which may be copyrighted. All PLOS content is published under the Creative Commons Attribution License (CC BY 4.0), which means that the manuscript, images, and Supporting Information files will be freely available online, and any third party is permitted to access, download, copy, distribute, and use these materials in any way, even commercially, with proper attribution. For these reasons, we cannot publish previously copyrighted maps or satellite images created using proprietary data, such as Google software (Google Maps, Street View, and Earth). For more information, see our copyright guidelines: http://journals.plos.org/plosone/s/licenses-and-copyright.

(1) You may seek permission from the original copyright holder of Figure 1 to publish the content specifically under the CC BY 4.0 license.  

Reviewers' comments:

Reviewer's Responses to Questions

**Comments to the Author**

1. Is the manuscript technically sound, and do the data support the conclusions?

Reviewer #1: No

Reviewer #2: No

Reviewer #3: Yes

2. Has the statistical analysis been performed appropriately and rigorously? 

Reviewer #1: No

Reviewer #2: Yes

Reviewer #3: Yes

3. Have the authors made all data underlying the findings in their manuscript fully available?

Reviewer #1: No

Reviewer #2: Yes

Reviewer #3: Yes

4. Is the manuscript presented in an intelligible fashion and written in standard English?

Reviewer #1: No

Reviewer #2: No

Reviewer #3: Yes

5. Review Comments to the Author

Reviewer #1: LN 153-154: Soil physical properties were measured by the cutting ring method. Q#: What is cutting ring method? There is no references? Which parameters were analysed by Cutting ring method?

LN 159-160: The HNO3-HCl-HClO4 atomic absorption spectrometry method was used to measure soil Ca and Mg. The contents of P, K, Na, Fe, Cu, Mn, Pb, Cd and Zn in soil were determined by ICP-AES.

Q#: Experimental methods – are extremely poor, cannot be accepted for any international journal.

Recommendation: REJECT.

Reviewer #2: 1.The language should be revised carefully. For example, 1) in the introduction part, Line 62"the Grain to Green Program (GTGP)" and Line 98 "the Grain-for-Green project" should be unified.2) and Line106, "three different types (CF, BD, EB) of forest vegetation (coniferous, evergreen broadleaved, broadleaved deciduous) "the abbreviations of CF, BD, EB are not corresponding with the order of the latter "coniferous, evergreen broadleaved, broadleaved deciduous". 3)And the abbreviations of Ks, SOC, SBD, et al., when they firstly show in the article, full name should be given.4) and the first paragraph is not so relevant with the subject of the article, it is suggested to be concise and more focused.

2. The methods should be more detailed, for example, 12 vegetations were selected to investigated, but the data you use to analyze conclude data of three types, so in the article you should tell the reader how you deal with the data.

3. About the chemical properties: Al, Ca, Cd, Cu, Fe, K, Mg, Mn, Na, P, Pb, and Zn, the relationship of the vegetation and these elements should be specified to explain the meaning of your study.

4.The biomass study is not relevant to your subject, or the relationship of the biomass study with the soil properties is not clearly explained in your article.

5.The conclusion "our results recommend planting the broadleaved deciduous species and coniferous species as the preferred tree species to enhance the soil fertility and water conservation functions", for your study is about the 12 vegetations dividedly, mixed forest is another story, your study can't support the conclusion.

So major revision is suggested.

Reviewer #3: Line 151-160 , Please supplement all references for soil physical and chemical properties determination methods.

Line 35-46,In the summary section, please add specific data for the related results. There are too many qualitative descriptions at this time.

Line 180-230，The results section has the problem of too many qualitative descriptions. Please add data about the results. For example，Line186，what is the highest value of the Liriodendron chinensis among 12 native species?

6. PLOS authors have the option to publish the peer review history of their article (what does this mean?). If published, this will include your full peer review and any attached files.

Reviewer #1: **Yes: **Prof Subodh Kumar Maiti

Reviewer #2: No

Reviewer #3: No

---

## [Author Response · Author response to Decision Letter 0]

10 Jan 2023

Responses to editor and reviewers

We highly appreciate the valuable comments and suggestions by the editor and reviewers on our manuscript. We have attempted to address each point. Some sentences have been revised or rewritten to improve the English. The following are our detail responses with reference to the order of the comments.

Part 1: Point-by-point response to Comments by the editor. 

Thank you for submitting your manuscript to PLOS ONE. After careful consideration, we feel that it has merit but does not fully meet PLOS ONE’s publication criteria as it currently stands. Therefore, we invite you to submit a revised version of the manuscript that addresses the points raised during the review process.

Re. Thanks for the positive comments. We believe this manuscript has important guiding significance in afforestation in karst area. And we revised the manuscript according to the editor and reviewers.

The study has value but the manuscript has some problems as suggested by the reviewers. The authors should respond to the comments of the reviewers one by one and revise the manuscript accordingly. The revised manuscript might be sent to the reviewers for further reviewing.

Re. Thanks a lot. And we revised the manuscript and respond to the comments of the reviewers one by one.

Re. Thanks for your suggestions. We have revised the file label according to the requirements.

Re. Thank you for reminding me. We have revised the financial disclosure. The specific information has been updated in our manuscript and cover letter. 

We look forward to receiving your revised manuscript.

Kind regards,

Jian Liu

Academic Editor

PLOS ONE

Journal Requirements:

Re. Thank you. We have revised the file name according to PLOS ONE's style requirements. 

Re. We are so sorry to make this mistake and we have revised the “Funding Information” and “Financial Disclosure”.

Re. Thanks! We have revised the grant numbers in the Funding Information section. 

Re. Thanks! And we have amended statements in our cover letter.

"NO authors have competing interests"

Re. Ok. We revised the state in this part.

Re. Ok. We will upload our study’s minimal underlying data set as either Supporting Information files.

Re. Ok. We did it as requested.

7. Please include a separate caption for each figure in your manuscript.

Re. Thanks! We have added the caption for each figure in our manuscript.

8. Please ensure that you refer to Figure 7 in your text as, if accepted, production will need this reference to link the reader to the figure.

Re. Thanks for your suggestion! We are so sorry to make this mistake. And we have corrected the figure number in our manuscript. 

9. Please upload a new copy of Figure 5 as the detail is not clear. Please follow the link for more information:

https://blogs.plos.org/plos/2019/06/looking-good-tips-for-creating-your-plos-figures-graphics/

https://blogs.plos.org/plos/2019/06/looking-good-tips-for-creating-your-plos-figures-graphics/

Re. Thanks for your advice. We have adjusted figure 5 to make it more clearly.

10. We note that Figure 1 in your submission contain map images which may be copyrighted. All PLOS content is published under the Creative Commons Attribution License (CC BY 4.0), which means that the manuscript, images, and Supporting Information files will be freely available online, and any third party is permitted to access, download, copy, distribute, and use these materials in any way, even commercially, with proper attribution. For these reasons, we cannot publish previously copyrighted maps or satellite images created using proprietary data, such as Google software (Google Maps, Street View, and Earth). For more information, see our copyright guidelines: http://journals.plos.org/plosone/s/licenses-and-copyright.

Re. Thanks for your reminder. We have removed the satellite images.

(1) You may seek permission from the original copyright holder of Figure 1 to publish the content specifically under the CC BY 4.0 license. 

Reviewers' comments:

Reviewer's Responses to Questions

Comments to the Author

1. Is the manuscript technically sound, and do the data support the conclusions?

Reviewer #1: No

Reviewer #2: No

Reviewer #3: Yes

2. Has the statistical analysis been performed appropriately and rigorously?

Reviewer #1: No

Reviewer #2: Yes

Reviewer #3: Yes

3. Have the authors made all data underlying the findings in their manuscript fully available?

Reviewer #1: No

Reviewer #2: Yes

Reviewer #3: Yes

4. Is the manuscript presented in an intelligible fashion and written in standard English?

Reviewer #1: No

Reviewer #2: No

Reviewer #3: Yes

5. Review Comments to the Author

Reviewer #1: LN 153-154: Soil physical properties were measured by the cutting ring method. Q#: What is cutting ring method? There is no references? Which parameters were analysed by Cutting ring method?

Re. Thanks for your questions. Sorry for the wrong expression about the method. The ring knife method is to use a ring knife with known mass and volume to cut the soil sample, weigh it and subtract the ring knife mass to obtain the mass of the soil. The volume of the ring knife is the volume of the soil, and then the density of the soil and other soil physical properties (field water capacity, capillary density) can be obtained.

Schematic diagram of the ring knife method

We have revised the Method part to make reader more clearly and we added the references to enhance the persuasion of our manuscript.

References:

Zhong FX, Xu XL, Li ZW, Zeng XM, Yi RZ, Luo W, Zhang YH, Xu CH. Relationships between lithology, topography, soil, and vegetation, and their implications for karst vegetation restoration. CATENA. 2022; 105831, 0341-8162. https://doi.org/10.1016/j.catena.2021.105831.

Blake GR, Hartge KH. Bulk density. In: Klute, A. (Ed.), Methods of Soil Analysis: Part 1, 2nd ed. America Society of Agronomy. 1986. Madison, pp. 363–375.

Gee GW, Bauder JW. Particle-size analysis. In: Klute, A. (Ed.), Methods of Soil Analysis. Part 1. Physical and Mineralogical Methods. American Society of Agronomy, 1986. Madison, pp. 383–411.

Again, sorry for the ambiguity expression in our manuscript.

LN 159-160: The HNO3-HCl-HClO4 atomic absorption spectrometry method was used to measure soil Ca and Mg. The contents of P, K, Na, Fe, Cu, Mn, Pb, Cd and Zn in soil were determined by ICP-AES.

Q#: Experimental methods – are extremely poor, cannot be accepted for any international journal.

Re. We are so sorry for the unclearly expression. We have revised the manuscript and added the references. We added the specific steps of the soil chemical properties experiment.

“Soil samples were decomposed by the HCL-HNO3 leaching method. Weigh 2.0g of air-dried soil sample in a triangular flask, then add 15 ml HCL (1 volume of HCL + 1 volume of distilled water) and 5 ml HNO3, add the plug and shake for 30 min, filter, fix the volume to 100ml, and waiting for measurement. A method to determine soil exchangeable potassium (K), calcium (Ca), sodium (Na), and magnesium (Mg) by using inductively coupled plasma (ICP-OES) (Plasma2000) and extraction with ammonium acetate was developed, which are measured by the Research Institute of Tropical Forestry, Chinese Academy of Forestry. The contents of P, Fe, Cu, Mn, Pb, Cd and Zn in soil were determined by ICP-AES”.

References:

Zhang YG, Xiao M, Dong YH, Jiang Y. Determination of soil exchangeable base cations by using atomic absorption spectrophotometer and extraction with ammonium acetate. Spectroscopy and Spectral Analysis. 2012. 32(8):2242-2245.

Lu RK. Soil agrochemical analysis methods [M]. Beijing: China Agricultural Science and Technology Press, 2000.

Recommendation: REJECT.

Reviewer #2: 1.The language should be revised carefully. For example, 1) in the introduction part, Line 62"the Grain to Green Program (GTGP)" and Line 98 "the Grain-for-Green project" should be unified.

Re. Thank you for your question. And we have revised the mistake.

2) and Line106, "three different types (CF, BD, EB) of forest vegetation (coniferous, evergreen broadleaved, broadleaved deciduous) "the abbreviations of CF, BD, EB are not corresponding with the order of the latter "coniferous, evergreen broadleaved, broadleaved deciduous".

Re. Thank you for your question. And we have revised the mistake.

3)And the abbreviations of Ks, SOC, SBD, et al., when they firstly show in the article, full name should be given.

Re. Thank you for your question. And we have revised the mistake.

4) and the first paragraph is not so relevant with the subject of the article, it is suggested to be concise and more focused.

Re. Thank you for your suggestions and we have revised the first paragraph to make the manuscript more focused.

2. The methods should be more detailed, for example, 12 vegetations were selected to investigated, but the data you use to analyze conclude data of three types, so in the article you should tell the reader how you deal with the data.

Re. Thank you for your advice. We have carefully modified the manuscript.

“In the present study, we divided 12 different tree species into three different forest vegetation types (CF, EB, BD). And we averaged the soil physical and chemical properties of three different forest vegetation types respectively for further data processing”.

3. About the chemical properties: Al, Ca, Cd, Cu, Fe, K, Mg, Mn, Na, P, Pb, and Zn, the relationship of the vegetation and these elements should be specified to explain the meaning of your study.

Re. Thanks for your question. We have added the chemical properties and vegetation relationship in our manuscript.

“Therefore, the study of differences in soil physicochemical properties, especially trace metal elements, among different vegetation restoration types is an important guideline for improving ecological restoration of natural and planted forests, especially in the subtropical karst region of southern China.”

4.The biomass study is not relevant to your subject, or the relationship of the biomass study with the soil properties is not clearly explained in your article.

Re. Thank you for your question. We removed the biomass part in our manuscript and focused on the difference in soil physical and chemical properties and the plant diversity during the vegetation restoration. 

5.The conclusion "our results recommend planting the broadleaved deciduous species and coniferous species as the preferred tree species to enhance the soil fertility and water conservation functions", for your study is about the 12 vegetations dividedly, mixed forest is another story, your study can't support the conclusion.

Re. Sorry for the inappropriate expression. And we have revised the conclusion part.

So major revision is suggested.

Reviewer #3: Line 151-160 , Please supplement all references for soil physical and chemical properties determination methods.

Re. Thanks for your suggestion. And we added the all references about soil physical and chemical properties determination methods.

References:

Zhong FX, Xu XL, Li ZW, Zeng XM, Yi RZ, Luo W, Zhang YH, Xu CH. Relationships between lithology, topography, soil, and vegetation, and their implications for karst vegetation restoration. CATENA. 2022; 105831, 0341-8162. https://doi.org/10.1016/j.catena.2021.105831.

Blake GR, Hartge KH. Bulk density. In: Klute, A. (Ed.), Methods of Soil Analysis: Part 1, 2nd ed. America Society of Agronomy. 1986. Madison, pp. 363–375.

Gee GW, Bauder JW. Particle-size analysis. In: Klute, A. (Ed.), Methods of Soil Analysis. Part 1. Physical and Mineralogical Methods. American Society of Agronomy, 1986. Madison, pp. 383–411.

Zhang YG, Xiao M, Dong YH, Jiang Y. Determination of soil exchangeable base cations by using atomic absorption spectrophotometer and extraction with ammonium acetate. Spectroscopy and Spectral Analysis. 2012. 32(8):2242-2245.

Lu RK. Soil agrochemical analysis methods [M]. Beijing: China Agricultural Science and Technology Press, 2000.

Line 35-46,In the summary section, please add specific data for the related results. There are too many qualitative descriptions at this time.

Re. Sorry for the inappropriate expression. And we have revised the summary section.

Line 180-230，The results section has the problem of too many qualitative descriptions. Please add data about the results. For example，Line186，what is the highest value of the Liriodendron chinensis among 12 native species?

Re. Sorry for the inappropriate expression. We added the specific data of the Liriodendron chinensis.

6. PLOS authors have the option to publish the peer review history of their article (what does this mean?). If published, this will include your full peer review and any attached files.

Do you want your identity to be public for this peer review? For information about this choice, including consent withdrawal, please see our Privacy Policy.

Reviewer #1: Yes: Prof Subodh Kumar Maiti

Reviewer #2: No

Reviewer #3: No

---

## [Decision Letter · Decision Letter 1]

13 Feb 2023

PONE-D-22-29404R1

Variation of soil physicochemical properties of different vegetation restoration types on subtropical karst area in southern China

PLOS ONE

Dear Dr. Ninghua,

Thank you for submitting your manuscript to PLOS ONE. After careful consideration, we feel that it has merit but does not fully meet PLOS ONE’s publication criteria as it currently stands. Therefore, we invite you to submit a revised version of the manuscript that addresses the points raised during the review process.

ACADEMIC EDITOR: In my opinion, the manuscript is much improved compared to the first draft. But as one reviewer pointed out, some problems still exist, the authors should respond to the comments and try to improve the quality of the manuscript.

We look forward to receiving your revised manuscript.

Kind regards,

Jian Liu

Academic Editor

PLOS ONE

Journal Requirements:

Reviewers' comments:

Reviewer's Responses to Questions

**Comments to the Author**

1. If the authors have adequately addressed your comments raised in a previous round of review and you feel that this manuscript is now acceptable for publication, you may indicate that here to bypass the “Comments to the Author” section, enter your conflict of interest statement in the “Confidential to Editor” section, and submit your "Accept" recommendation.

Reviewer #2: (No Response)

Reviewer #4: All comments have been addressed

2. Is the manuscript technically sound, and do the data support the conclusions?

Reviewer #2: No

Reviewer #4: Yes

3. Has the statistical analysis been performed appropriately and rigorously? 

Reviewer #2: No

Reviewer #4: Yes

4. Have the authors made all data underlying the findings in their manuscript fully available?

Reviewer #2: No

Reviewer #4: (No Response)

5. Is the manuscript presented in an intelligible fashion and written in standard English?

Reviewer #2: Yes

Reviewer #4: Yes

6. Review Comments to the Author

Reviewer #2: 1.The method that the authors dealt with the data is not right. "In the present study, we divided 12 different tree species into three different forest vegetation types (CF, EB, BD). And we averaged the soil physical and chemical properties of three different forest vegetation types respectively for further data processing”.

2.About the chemical properties: Al, Ca, Cd, Cu, Fe, K, Mg, Mn, Na, P, Pb, and Zn, the relationship of the vegetation and these elements is still not clear in the study.

3.There is still something wrong in the article, for example, "The soil chemical and physical properties and plant diversity of 10 vegetation types in Northwest Hunan were analysed."

4.The conclusion is not sound because all the analyzing is based on the three group CF, EB, BD, but not "Liriodendron chinense (Hemsl.) Sarg".

Reviewer #4: The manuscript is much improved compared to the first draft. The authors did well to add experimental and analytical details, improving the clarity of their writing. I think the manuscript is close to publication-ready.

7. PLOS authors have the option to publish the peer review history of their article (what does this mean?). If published, this will include your full peer review and any attached files.

Reviewer #2: No

Reviewer #4: No

---

## [Author Response · Author response to Decision Letter 1]

17 Feb 2023

Responses to editor and reviewers

We highly appreciate the valuable comments and suggestions by reviewers on our manuscript. We made further modifications according to the suggestions of the reviewers and have attempted to address each point. The following are our details responses with reference to the order of the comments.

Part 1: Response to Comments by the editor. 

PONE-D-22-29404R1

Variation of soil physicochemical properties of different vegetation restoration types on subtropical karst area in southern China

PLOS ONE

Dear Dr. Ninghua,

Thank you for submitting your manuscript to PLOS ONE. After careful consideration, we feel that it has merit but does not fully meet PLOS ONE’s publication criteria as it currently stands. Therefore, we invite you to submit a revised version of the manuscript that addresses the points raised during the review process.

ACADEMIC EDITOR: In my opinion, the manuscript is much improved compared to the first draft. But as one reviewer pointed out, some problems still exist, the authors should respond to the comments and try to improve the quality of the manuscript.

Re. Thanks for your positive comments on our manuscript. The suggestions put forward by the reviewers are very helpful for the further optimization of our paper. We make point-to-point responses based on the comments of reviewers.

Part 1: Point-by-point response to Comments by the reviewer. 

Comments to the Author

1. If the authors have adequately addressed your comments raised in a previous round of review and you feel that this manuscript is now acceptable for publication, you may indicate that here to bypass the “Comments to the Author” section, enter your conflict of interest statement in the “Confidential to Editor” section, and submit your "Accept" recommendation.

Reviewer #2: (No Response)

Reviewer #4: All comments have been addressed

Re. Thanks for your comments

2. Is the manuscript technically sound, and do the data support the conclusions?

Reviewer #2: No

Reviewer #4: Yes

Re. Thanks for your comments

3. Has the statistical analysis been performed appropriately and rigorously?

Reviewer #2: No

Reviewer #4: Yes

Re. Thanks for your comments

4. Have the authors made all data underlying the findings in their manuscript fully available?

Reviewer #2: No

Reviewer #4: (No Response)

Re. Thanks for your comments

5. Is the manuscript presented in an intelligible fashion and written in standard English?

Reviewer #2: Yes

Reviewer #4: Yes

Re. Thanks for your comments

6. Review Comments to the Author

Reviewer #2: 

1.The method that the authors dealt with the data is not right. "In the present study, we divided 12 different tree species into three different forest vegetation types (CF, EB, BD). And we averaged the soil physical and chemical properties of three different forest vegetation types respectively for further data processing”.

Re. Sorry for the wrong expression. We have revised this paragraph as “In this study, soil physical and chemical properties of 12 tree species were treated on average (groups of repetitions), and then 12 different tree species were divided into three different forest vegetation types (CF, EB, BD) for further data processing.” Hope the revised version could be more clear.

2.About the chemical properties: Al, Ca, Cd, Cu, Fe, K, Mg, Mn, Na, P, Pb, and Zn, the relationship of the vegetation and these elements is still not clear in the study.

Re. Sorry. We have added an analysis of the relationship between the vegetation and these elements, which is shown in the part “Effects of stand type characteristics on soil basic physical and chemical properties”. 

3.There is still something wrong in the article, for example, "The soil chemical and physical properties and plant diversity of 10 vegetation types in Northwest Hunan were analysed."

Re. Sorry for the mistakes. We have rechecked the manuscript and made further modifications.

4.The conclusion is not sound because all the analyzing is based on the three group CF, EB, BD, but not "Liriodendron chinense (Hemsl.) Sarg".

Re. Thanks for your valuable comments. We have revised this part as “For these characteristics, the broadleaved deciduous forest may be the most suitable forest type in forest restoration management in the subtropical karst area of southern China.”

Reviewer #4: The manuscript is much improved compared to the first draft. The authors did well to add experimental and analytical details, improving the clarity of their writing. I think the manuscript is close to publication-ready.

Re. Thanks for your positive comments.

---

## [Editor Report · Decision Letter 2]

21 Feb 2023

Variation of soil physicochemical properties of different vegetation restoration types on subtropical karst area in southern China

PONE-D-22-29404R2

Dear Dr. Ninghua,

We’re pleased to inform you that your manuscript has been judged scientifically suitable for publication and will be formally accepted for publication once it meets all outstanding technical requirements.

Kind regards,

Jian Liu

Academic Editor

PLOS ONE
---

## [Editor Report · Acceptance letter]

6 Mar 2023

PONE-D-22-29404R2 

Variation of soil physicochemical properties of different vegetation restoration types on subtropical karst area in southern China 

Dear Dr. Zhu:

I'm pleased to inform you that your manuscript has been deemed suitable for publication in PLOS ONE. Congratulations! Your manuscript is now with our production department. 

Kind regards, 

on behalf of

Dr. Jian Liu 

Academic Editor

PLOS ONE